# Transthoracic Impedance Measured with Defibrillator Pads—New Interpretations of Signal Change Induced by Ventilations

**DOI:** 10.3390/jcm8050724

**Published:** 2019-05-22

**Authors:** Per Olav Berve, Unai Irusta, Jo Kramer-Johansen, Tore Skålhegg, Håvard Wahl Kongsgård, Cathrine Brunborg, Elisabete Aramendi, Lars Wik

**Affiliations:** 1National Advisory Unit for Prehospital Emergency Medicine (NAKOS), Oslo University Hospital—Ullevål and University of Oslo, P.O. Box 4956 Nydalen, N-0424 Oslo, Norway; jo.kramer-johansen@medisin.uio.no (J.K.-J.); haavard.kongsgaard@gmail.com (H.W.K.); 2Prehospital Clinic, Ambulance Department, Oslo University Hospital—Ullevål, P.O. Box 4956 Nydalen, N-0424 Oslo, Norway; toretang911@hotmail.com; 3Communications Engineering Department, University of the Basque Country UPV/EHU, Alameda Urquijo S/N, 48013 Bilbao, Spain; unai.irusta@ehu.eus (U.I.); elisabete.aramendi@ehu.eus (E.A.); 4Prehospital Clinic, Airambulance Department, Oslo University Hospital—Ullevål, P.O. Box 4956 Nydalen, N-0424 Oslo, Norway; 5Oslo Centre for Biostatistics and Epidemiology, Research Support Services, Oslo University Hospital—Ullevål, P.O. Box 4956 Nydalen, N-0424 Oslo, Norway; UXBRUC@ous-hf.no

**Keywords:** transthoracic impedance, ventilation, peak inspiration pressure, cardiopulmonary resuscitation (CPR), pulmonary injury, pulmonary barotrauma, ventilation pattern

## Abstract

Compressions during the insufflation phase of ventilations may cause severe pulmonary injury during cardiopulmonary resuscitation (CPR). Transthoracic impedance (TTI) could be used to evaluate how chest compressions are aligned with ventilations if the insufflation phase could be identified in the TTI waveform without chest compression artifacts. Therefore, the aim of this study was to determine whether and how the insufflation phase could be precisely identified during TTI. We synchronously measured TTI and airway pressure (Paw) in 21 consenting anaesthetised patients, TTI through the defibrillator pads and Paw by connecting the monitor-defibrillator’s pressure-line to the endotracheal tube filter. Volume control mode with seventeen different settings were used (5–10 ventilations/setting): Six volumes (150–800 mL) with 12 min^−1^ frequency, four frequencies (10, 12, 22 and 30 min^−1^) with 400 mL volume, and seven inspiratory times (0.5–3.5 s) with 400 mL/10 min^−1^ volume/frequency. Median time differences (quartile range) between timing of expiration onset in the Paw-line (Paw_EO_) and the TTI peak and TTI maximum downslope were measured. TTI peak and Paw_EO_ time difference was 579 (432–723) ms for 12 min^−1^, independent of volume, with a negative relation to frequency, and it increased linearly with inspiratory time (slope 0.47, $R^{2}$ = 0.72). Paw_EO_ and TTI maximum downslope time difference was between −69 and 84 ms for any ventilation setting (time aligned). It was independent ($R^{2}$ < 0.01) of volume, frequency and inspiratory time, with global median values of −47 (−153–65) ms, −40 (−168–68) ms and 20 (−93–128) ms, for varying volume, frequency and inspiratory time, respectively. The TTI peak is not aligned with the start of exhalation, but the TTI maximum downslope is. This knowledge could help with identifying the ideal ventilation pattern during CPR.

## 1. Introduction

International guidelines for cardiopulmonary resuscitation (CPR) recommend specific treatment goals for chest compressions and ventilations [1]. Noninvasive measurements of regional chest impedance can be used to monitor chest compression and ventilation activity. In an intensive care unit, advanced technologies such as electrical impedance tomography may be available to monitor ventilations [2,3]. However, in an emergency prehospital setting the availability of technology is limited, and most defibrillators only incorporate circuitry to measure the resistance to the alternating current between the defibrillator pads, i.e., the transthoracic impedance (TTI) [4,5,6]. Signal processing algorithms exist to identify compressions and ventilations in the TTI, which can therefore be used to monitor and evaluate these two CPR events, and the CPR quality metrics derived thereof [7,8,9,10,11,12].

Ventilations during CPR are provided with different strategies, even during continuous chest compressions in intubated patients performed in compliance with treatment recommendations. Manual or mechanical ventilations can for example be given as brief ventilations interposed between compressions [13], or as slower ventilations impacting several compressions [14,15,16,17]. In either case no clear definition is set for which tidal volume or peak airway pressure is considered advantageous or safe. Differing ventilation patterns during CPR are likely to cause different amounts of collisions between ventilations and chest compressions. Such collisions can generate short bursts of very high intrathoracic pressures that may cause pulmonary barotrauma [14,18,19,20,21,22]. This effect has been suggested to be more pronounced in the inspiratory phase of positive pressure ventilation, when airway pressure increases and the expiration valve is closed [23]. A chest compression provided after the start of exhalation may cause less increase in pressure, as gas has an escape route out of the airway [14,18].

A proper analysis of ventilation and compression collisions should therefore identify which phase of the ventilation a collision occurs. Such analysis would be considerably facilitated by a reliable method to discriminate the inspiratory from the expiratory phase in the TTI waveform, since TTI is the only signal during CPR where both compression and ventilation activities are concurrently visible. The peak of the change induced in the TTI curve by ventilations (TTI peak) has been customarily regarded as the onset of expiration [9,24]. However, most ventilation induced changes in the TTI signal have a differing shape from the corresponding airway pressure (Paw) and tidal volume curves. Therefore, it is unclear if the TTI peak accurately identifies the time point representing the end of inspiration and the beginning of exhalation in the Paw curve (Paw_EO_), when airway pressures and volumes start to decline.

Our aims for the present study, conducted in a controlled ventilation setting of anesthetized adult patients in the operating room (OR), were first to investigate if the TTI peak corresponds with the start of expiration, and then to identify if other features of the TTI signal better discriminate inspiration from expiration. Finally, we investigated how these TTI curve points are affected by tidal volume, inspiratory time, ventilation frequency and patient chest dimensions.

## 2. Materials and Methods

### 2.1. Study Design and Patient Recruitment

Adult patients scheduled for elective surgery in general anesthesia at Oslo University hospital (OUH), location Ullevål, were eligible for inclusion. The ethical committee on medical scientific research of South-Eastern Norway approved the study (2015/1287). Patients gave written consent after oral and written information. One of the investigators provided this information and collected the consent during the preoperative meeting at least one day before scheduled surgery. All patients could withdraw their consent at any time without consequences for scheduled surgery or care, as per ethical regulations. Additional patient data such as age, gender, height, weight or chest circumference were added to the de-identified patient record, and were later used to derive body mass index (BMI). Patients with chronic obstructive pulmonary disease, severe heart failure, severe kidney failure, and raised intracranial pressure were not recruited. Inclusion was decided by the investigators in close communication with the anesthesiologist in charge of the OR.

In the OR, two investigators not involved in the clinical treatment of the patient attached standard defibrillator pads in the anterior-lateral position to the LifePak 15 (LP15) monitor-defibrillator (Physio-Control Inc., Redmond, WA, USA). Electrocardiogram (ECG) and TTI were recorded through the defibrillator pads, and TTI was obtained in the LP15 by injecting a sinewave excitation current (20 kHz and 34 μA) and measuring the voltage drop between the electrodes. The LP15 invasive pressure line was used to monitor the timing of the Paw change by connecting the fluid filled line directly to a port on the airway filter attached to the endotracheal tube. This setup has been used in several experimental settings [25,26], and enabled the LP15 monitor to simultaneously record the timing of change of Paw and TTI signals. Paw readings were presented in real-time on the LP15 monitor. The precision of the timing of the Paw change was evaluated by concurrent video recording of the ventilator and the LP15 monitor screen at 240 frames per second (fps) using the GoPro Hero 3 camera. This setup allowed a precise assessment of the delay between the Paw line seen in the LP15 and the actual ventilator waveforms. All other instrumentation and monitoring were unchanged and in accordance with the directions by the anesthesiologist in charge.

### 2.2. Experimental Protocol

After induction of general anesthesia with the use of a muscle relaxant (cisatracurium), both the anesthetist in charge and the investigators double-checked if the endotracheal tube was correctly positioned and that airway pressures, respiratory rate, pETCO_2_ and oxygenation were kept within the desired range, with minor differences between patients due to clinical considerations of the anesthesiologist in charge of patient treatment. Then we provided normal ventilation for at least 5 min ensuring adequate minute volume and pETCO_2_. Two ventilators were used depending on OR setup, the LEON+ (Heinen-Löwenstein, Avalon, Germany) and the Primus^®^ (Dräger, Germany).

All ventilations in this experiment were provided with volume control mode, with no positive end-expiratory pressure. Volume-controlled ventilations allowed the provision of precise volumes in a stable manner, and the reaching of target volumes within a short time frame. Furthermore, since high flow and increasing pressures were delivered until a certain volume was reached, the volume-controlled mode best resembled the use of manual ventilations in most critical care situations. The experiment started with six series of five ventilations with inspiration to expiration ratio (I:E ratio) of 1:2, at a fixed respiratory rate of 12 min^−1^, but with different volumes of 150, 200, 300 (10 patients), 400, 600 and 800 mL. Then, the volume was fixed at 400 mL to deliver four series of 10 ventilations with I:E ratio 1:2 at these frequencies: 10, 12, 22 and 30 min^−1^. Finally, keeping volume and frequency fixed at 400 mL and 10 min^−1^, seven series of 10 ventilations were delivered with these inspiratory times: 0.5, 1.0, 1.5, 2.0, 2.5, 3.0 and 3.5 s.

### 2.3. Measurement of Variables of Interest

We replayed the video recordings frame by frame to count the number of frames between the same event on the LP15 and on the ventilator screen. We compared the timing of the Paw_EO_ seen in the LP15 to the timing of the drop of pressure, flow and volume that marked the start of exhalation on the ventilator screens. We annotated this delay manually for each of the ventilator setups to confirm that TTI and Paw were synchronously recorded by the LP15.

The LP15 data from CODE-STAT (Physio-Control Inc., Redmond, WA, USA) was converted into comma-separated values files, and imported into a custom made Matlab^®^ (Mathworks Inc., Natick, MA, USA) application for data visualization and annotation. The time resolution of the data was 10 ms, i.e., the sampling rate was 100 samples per second. We processed the TTI signal to remove the circulation component, applying the algorithms conceived to detect pulse in cardiac arrest through the defibrillation pads [27]. Band pass filtering (0.7–7 Hz) of the TTI signal removed high frequency noise and low frequency baseline fluctuations, and resulted in the ventilation TTI waveform, as shown in Figure 1. All filters were used in a forward-backward configuration which ensured that no time delay was introduced by filtering. Our custom made Matlab tool automatically detected the onset, peak and offset values of the ventilations [9,28] in the TTI and Paw lines (Figure 1). Two researchers manually audited the automatic annotations until a consensus was reached. The Paw line was used to detect the timing of the airway pressure change, and to enable real-time data collection of pressure change timing. The timing of inspiration onset and expiration onset were correctly identified using this setup because they produced marked increases/decreases in the pressure readings of the LP15 invasive pressure line (see Figure 1).

Once data was (semi)-automatically annotated an algorithm automatically marked the TTI maximum upslope and maximum downslope in the TTI ventilation waveform. The time differences between the TTI peak, TTI maximum downslope and Paw_EO_ were analyzed in terms of volume (six series), frequency (four series) and inspiratory time (seven series).

### 2.4. Statistical Analysis

Sample size estimation was performed using the difference in time (seconds) of the start of ventilations registered in the LP15 Paw line and the peak of the corresponding TTI signal. With an estimated mean difference of 0.5 s between start and end, and an estimated standard deviation of 0.75 s, considering a type I error of 5% and power of 90% we would need to include 20 patients (paired samples).

Time differences for each ventilator setting were tested for normality using the Kolmogorov–Smirnov normality test. Since 23 of the 34 series (17 settings, two time differences) did not pass the normality test, we report the data as median and with a 25–75 percentile range. The 95 % confidence interval of the median delay for each setting was estimated using nonparametric confidence intervals for quantiles [29]. For the relationship between time differences and volume, inspiratory time and frequency, data was fitted using a linear regression model of the form $t\;∽\;\beta_{0}+\beta_{1}x$, where *t* is the time difference, $\beta_{1}$ and $\beta_{0}$ the slope and the intercept, and *x* stands for the independent variable. The coefficient of determination ($R^{2}$) for each relation provides an estimate of how well the independent variables explained the observed variation. A generalized linear mixed model was used to account for repeated measures, since 5–10 ventilations per patient were analyzed for each ventilator setting. Differences in median time differences between small (<97.5cm) and large chest (≥97.5cm) and normoweight (BMI < 25) and overweight (BMI ≥ 25) patients were assessed using the Wilcoxon Rank Sum test for independent group comparisons, and a two tailed *p* value < 0.01 was considered significant.

## 3. Results

A total of 21 patients (nine female) were enrolled in the study, from which 12 (six female) were normoweight and nine (three female) overweight patients, and eight with small and eight with large chest (data on chest circumference was missing for five patients). Airway pressure curve changes were visualized earlier in the Paw curve on the LP15 than in the corresponding flow, volume and pressure curves on the ventilator monitor. This difference was less than 1–2 picture frames at 240 fps (maximum 8 ms) for Paw inspiration onset, and 3–4 picture frames (maximum 16 ms) for Paw_EO_ readings. These differences were consistent throughout the experiment.

Figure 2, Figure 3 and Figure 4 show the time difference between the TTI fiducial points and the peak inspiration pressure time point in the Paw curve. Paw_EO_ was not time-aligned with the TTI peak for any ventilator setting (Table 1), although the time difference was small for very short inspiratory times (Figure 3) and high ventilation rates (Figure 4). As shown in Table 1, the 95 % confidence intervals for the median time differences presented a minimum lower bound of 560 ms, 250 ms and 230 ms for the experiments with controlled volume (Figure 2), inspiratory time (Figure 3) and frequency (Figure 4), respectively. The time difference between the TTI peak and Paw_EO_ was independent of tidal volume (slope $-1.0\times 10^{-5}$ mL, $R^{2}$ < 0.01) and showed a linear positive relation with inspiratory time with a slope of 0.47 (no units, $R^{2}=0.72$), and a negative relation with ventilation frequency (Figure 4). Time differences were very large, above 1.5 s, for very long inspiratory times of 3 s and above.

The TTI maximum downslope was well aligned with Paw_EO_ independently of tidal volume (Figure 2), inspiratory time (Figure 3) or ventilation frequency (Figure 4). As shown in Table 1, the largest median time difference was −69 ms, 84 ms and −60 ms for the experiments with controlled volume (Figure 2), inspiratory time (Figure 3) and frequency (Figure 4), respectively. Unlike for TTI peak, the time differences with TTI maximum downslope were independent of volume (slope $7.8\times 10^{-5}\;s/\mathrm{mL}$, $R^{2}$ < 0.01), inspiratory time (slope $1.9\times 10^{-2}$, $R^{2}$ < 0.01) and frequency (slope $-1.3\times 10^{-3}\;s/\min-1$, $R^{2}$ < 0.01). These time differences showed no significant differences for different chest sizes or normo/overweight patients except for five settings (out of 34 comparisons) with no discernible pattern (Table 2).

## 4. Discussion

Our data shows that the maximum TTI peak is not time aligned with the onset of expiration measured via airway pressure recordings (Paw_EO_) in a controlled experiment in anesthetized patients. The TTI peak occurs before the expiration valve opens, and thus before the abrupt deflation of airway volumes and reduction in airway pressures that define start of expiration in the ventilator volume, pressure and flow lines. This was precisely represented in our setup via the Paw_EO_. The time difference between the TTI peak and onset of expiration increased linearly with insufflation times, this suggests that the peak in the TTI ventilation waveform is caused by other mechanisms than volume change alone.

Interestingly, we found that the maximum downslope of the TTI ventilatory waveform consistently corresponded to the Paw_EO_, a finding that was reproduced for a wide range of tidal volumes, inspiratory times and ventilation frequencies. These time differences were independent of the chest or weight group of the patient, and were thus not associated to chest physiognomy or dimensions. These new findings need further confirmation, especially in the CPR setting. TTI during CPR shows both activity due to chest compressions and to ventilations [7,9], and the extraction of the ventilation waveform may be difficult. In most cases a visual assessment of the maximum downslope point will not be possible and signal processing techniques would have to be applied to the TTI to obtain ventilation waveforms. Those techniques are similar to the ones used to remove the circulation component from our data (see Figure 1) [27], although chest compression activity has larger amplitudes. The methods developed to remove chest compression artefacts from the ECG could be adapted to remove chest compression activity from the TTI, both for manual CPR [30] and mechanical CPR [31]. This would open the possibility of analyzing different ventilation/compression strategies during CPR, for example by comparing compression/ventilation collision data to information on outcome and chest injuries.

These findings are relevant because the mechanism causing pulmonary barotrauma is still unknown [19]. It occurs in various degrees during CPR and can result in a range of deleterious pathophysiological effects like pulmonary contusion, edema [20], atelectasis, bleeding and pneumothorax, leading to reduced gas exchange and possibly to impaired circulation [21]. Pulmonary tissue damage is considered to be an important contributor to post cardiac arrest syndrome, the inflammatory sepsis-like state that often affects patients after return of spontaneous circulation [32].

The misalignment between TTI peak and Paw_EO_ could be due to the combined effects of positive pressure ventilation on gas content (poor electricity conductor) of the pulmonary tissue and on pulmonary blood flow (good conductor). Electrodes were correctly placed in the antero-lateral position, as certified by the investigators in the OR. In this configuration the ventilatory component of the impedance is affected mostly by changes in the anterior non-dependent regions of pulmonary tissue. The pulmonary blood flow is low in this region in supine positive pressure ventilated patients [33]. Small increases in intrathoracic pressures will lead to filling of the anterior alveoli first, as they are already kept open by gravity and thus need the lowest opening and filling pressures. Most of the delivered volumes will therefore be located in these anterior parts of the chest. At the same time, increased airway pressures squeeze blood away from the highest, most anterior regions of the lung at an early stage of inspiration. The increased gas and reduced blood content produce the increase in the impedance represented as the upstroke part of the TTI signal. During the end of the inspiratory phase, pooling of blood on the arterial side of the pulmonary circulation may increase pulmonary artery pressures, forcing blood to reenter the anterior pulmonary vascular bed. This would explain the reduction in thoracic impedance despite maintained or even increasing intrathoracic gas volumes.

During expiration this phenomenon is reversed. When the intrathoracic pressure drops, the highest volume change per unit time occurs anteriorly. It is also likely that an increased return of blood to anterior parts of the lung occurs simultaneously. Thus, the steepest reduction in TTI, the maximum downslope point, would occur soon after the ventilator opens the expiratory valve to allow expiration. In our study the time points of the maximum TTI downslope and Paw_EO_ were tightly aligned over a wide range of tidal volumes, inspiratory times and ventilation frequencies.

Precise quantification of ventilation/compression collisions allows for further studies into the nature of lung injury during CPR. Injuries and deleterious side effects of CPR are common concerns among lay rescuers and professionals, and possibilities to relate these to more precise quantification of collisions between chest compressions and ventilations can provide better evidence to inform our teaching and practice regarding ventilation pattern. Our findings suggest that it could be possible to use the time point of the TTI maximum downslope in a future algorithm to guide safe provision of chest compressions, as this seems to indicate the first sign of an open airway.

### Limitations

Even if TTI is commonly used for detection of ventilations [9,34] and compressions [7], CPR quality reporting using TTI has lower accuracy when compared to other technologies like capnography for ventilations [35,36], or accelerometers or force equipment for compressions [28]. Compression depth and ventilation volume cannot be measured using TTI [34,37]. Furthemore, in some cases manufacturers do not disclose the precise filters and algorithms used to smooth and enhance TTI signals.

We also discovered that the Paw line changes consistently occurred earlier in the Paw curve presentation on the LP15 screen (1–4 picture frames at 240 fps) than on the ventilator screen. Although the time difference is negligible (≤16 ms) it could be caused by the software of monitoring equipment that delays the signal on the screen. The custom made airway pressure monitoring setup was connected as close to the patient as possible. This could explain why expiration occurred a bit sooner on the LP15 screen compared to the ventilator monitor that presents measurements done further away from the patient.

CPR was not provided during this study. In order to draw parallels into effects during CPR, these effects must be investigated in prospective studies of CPR.

## 5. Conclusions

Transthoracic impedance peak readings do not precisely identify the start of expiration in anesthetized patients with positive pressure ventilation. The time difference between the TTI peak and the start of expiration varies with inspiratory time and frequency, but not with tidal volume.

The maximum downslope of the ventilation TTI curve is time aligned with the start of expiration independently of tidal volume, inspiratory time and ventilation frequency. This could be relevant in future investigations on the relationship between chest compressions and ventilations during CPR. Further research is needed to confirm these findings in human CPR settings.

## Figures and Tables

**Figure 1 jcm-08-00724-f001:**
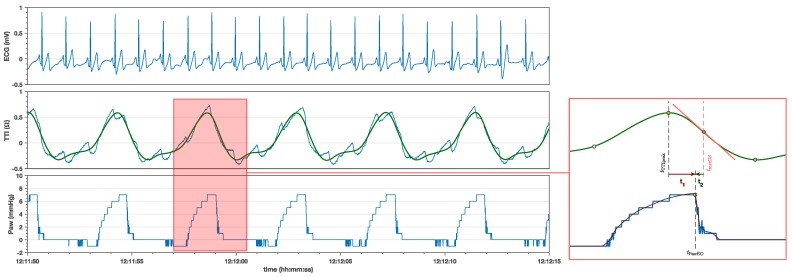
A 25 s example of the data recorded by the LP15 with the electrocardiogram (ECG) (top channel), raw transthoracic impedance (TTI) (middle channel) and airway pressure (Paw) line (bottom channel). The raw impedance (blue) shows a strong circulation component aligned with the QRS complexes in the ECG. This component was filtered to obtain the ventilatory component in green. The red square highlights a typical ventilation waveform and the fiducial points in the TTI. As reference an interpolated Paw curve (in red) is superposed to the Paw curve (blue) to show how Paw gradually approaches the ${\mathrm{Paw}}_{\mathrm{EO}}$ point. The time intervals $t_{1}=t_{\mathrm{PawEO}}-t_{\mathrm{TTIpeak}}$ and $t_{2}=t_{\mathrm{PawEO}}-t_{\mathrm{maxDS}}$ are the ones used in Figure 2, Figure 3 and Figure 4.

**Figure 2 jcm-08-00724-f002:**
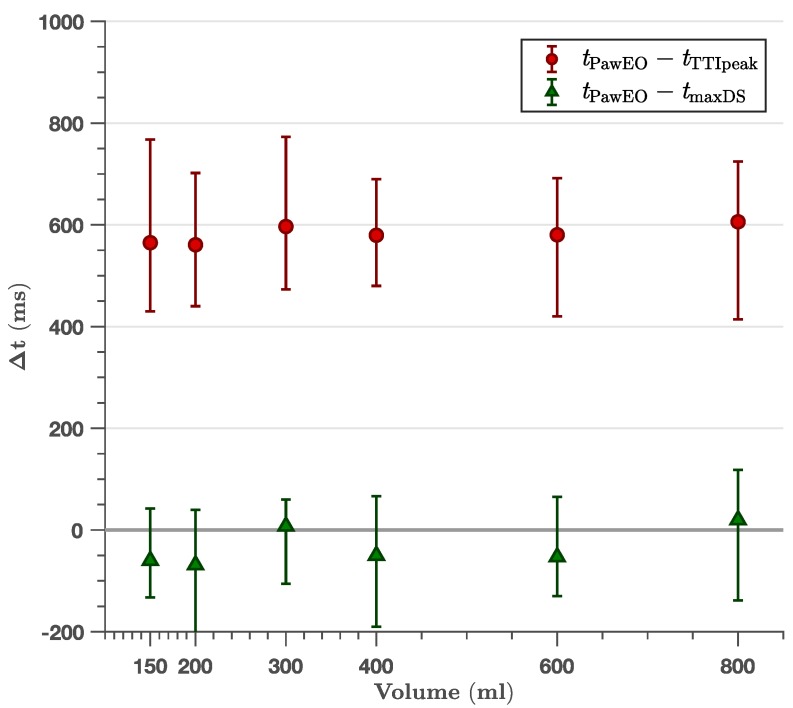
Effect of volume on the time differences between the TTI fiducial points (TTI peak, $t_{\mathrm{TTIpeak}}$, and maximum downslope, $t_{\mathrm{maxDS}}$) and the peak pressure time in the Paw line marking exhalation onset ($t_{\mathrm{PawEO}}$). All ventilator settings had a constant frequency of 12 min^−1^. Data is shown as median with first to third quartile interval. Time differences were independent of volume ($R^{2}$ < 0.01 for the linear fit), and were around 0.5 s for TTI peak and around 0 s for maximum downslope.

**Figure 3 jcm-08-00724-f003:**
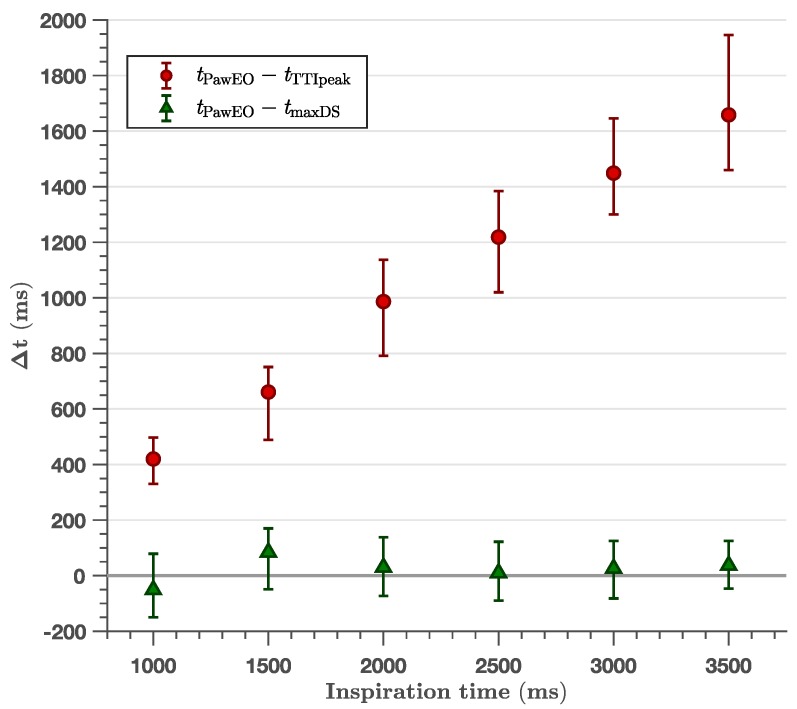
Effect of inspiratory time in the time differences between the TTI fiducial points (TTI peak, $t_{\mathrm{TTIpeak}}$, and maximum downslope, $t_{\mathrm{maxDS}}$) and the peak pressure time in the Paw line marking exhalation onset ($t_{\mathrm{PawEO}}$). All ventilator settings had a constant tidal volume of 400 mL and frequency of 10 min^−1^. Data is shown as median with first to third quartile interval. There was a linear relation with positive slope 0.47 for the TTI peak, and no relation with a median time difference for maximum downslope ($R^{2}$ < 0.01).

**Figure 4 jcm-08-00724-f004:**
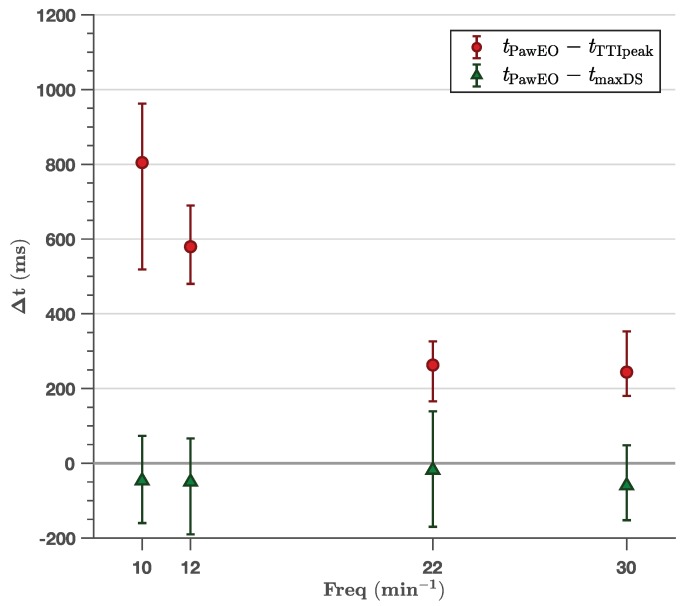
Effect of ventilation frequency in the time differences between the TTI fiducial points (TTI peak, $t_{\mathrm{TTIpeak}}$, and maximum downslope, $t_{\mathrm{maxDS}}$) and the peak pressure time in the Paw line marking exhalation onset ($t_{\mathrm{PawEO}}$). All ventilator settings had a constant tidal volume of 400 mL. The time difference for TTI peak decreased as frequency increased, although not linearly. There was no relation with a median time difference for maximum downslope ($R^{2}$ < 0.01).

**Table 1 jcm-08-00724-t001:** Median time differences between Paw_EO_ and TTI peak ($t_{1}=t_{{\mathrm{Paw}}_{\mathrm{EO}}}-t_{\mathrm{TIpeak}}$) and Paw_EO_ and TTI maximum downslope point ($t_{2}=t_{{\mathrm{Paw}}_{\mathrm{EO}}}-t_{\mathrm{maxDS}}$) for all the ventilator settings. The values are shown as median and 95% confidence intervals.

Ventilator Mode	Time Differences	
**vol (mL)/freq (min^−1^)/$t_{i}$ (ms)**	**$t_{1}$ (ms)**	**$t_{2}$ (ms)**
**varying vol**		
150/12/2000	565 (514, 648)	−60 (−89, −18)
200/12/2000	561 (542, 596)	−69 (−127, −8)
300/12/2000	597 (558, 711)	8 (−67, 48)
400/12/2000	580 (542, 609)	−50 (−98, 8)
600/12/2000	580 (540, 636)	−53 (−83, −20)
850/12/2000	606 (571, 628)	20 (−52, 56)
**varying $t_{i}$**		
400/10/ 500	263 (248, 279)	−18 (−40, 4)
400/10/1000	420 (402, 434)	−51 (−91, −7)
400/10/1500	661 (631, 674)	84 (30, 113)
400/10/2000	987 (969, 1049)	30 (9, 50)
400/10/2500	1219 (1200, 1237)	10 (−10, 39)
400/10/3000	1449 (1394, 1496)	26 (−4, 52)
400/10/3500	1658 (1584, 1831)	37 (10, 70)
**varying freq**		
400/10/2000	805 (744, 899)	−47 (−82, −23)
400/12/2000	580 (542, 609)	−50 (−98, 8)
400/22/2000	263 (248, 279)	−18 (−40, 4)
400/30/2000	244 (226, 270)	−60 (−76, −20)

**Table 2 jcm-08-00724-t002:** Median time differences with 95 % CI between Paw_EO_ and TTI maximum downslope for normoweight patients (BMI < 25, 12 patients) and overweight patients (BMI ≥ 25, nine patients). There were no significant differences (at the 0.01 level) between normoweight and overweight patients except at three ventilator settings (indicated by ${}^{†}$) with no discernible pattern.

Ventilator Mode	$t_{2}$ (ms)		*p*-Value	$t_{2}$ (ms)		*p*-Value
vol/freq/$t_{i}$	BMI < 25	BMI ≥ 25		Chest < 97.5 cm	Chest ≥ 97.5 cm	
**varying vol**						
150/12/2000	−24 (−70, 20)	−90 (−130, −29)	0.02	−77 (−143, −22)	−70 (−108, 13)	0.44
200/12/2000	−45 (−141, 7)	−85 (−179, 18)	0.37	−190 (−259, −100)	−58 (−100, 14)	<0.01 ${}^{†}$
300/12/2000	49 (−10, 120)	−77 (−142, 41)	<0.01 ${}^{†}$	−136 (−185, 47)	−49 (−72, 45)	0.08
400/12/2000	−50 (−84, 6)	−53 (−120, 31)	0.57	−161 (−210, −41)	−67 (−110, 14)	0.36
600/12/2000	−48 (−83, −4)	−72 (−110, 11)	0.72	−70 (−170, −15)	−64 (−98, 16)	0.45
850/12/2000	−14 (−52, 52)	32 (−80, 91)	0.77	−30 (−139, 61)	−10 (−67, 59)	0.89
**varying $t_{i}$**						
400/10/ 500	50 (−152, 107)	−20 (−46, −9)	0.39	−1 (−20, 82)	−10 (−195, 75)	0.15
400/10/1000	−73 (−99, −49)	10 (−102, 57)	0.64	−125 (−140, −64)	−29 (−66, 32)	0.13
400/10/1500	77 (29, 100)	121 (0, 153)	0.21	−40 (−70, 8)	140 (100, 170)	<0.01 ${}^{†}$
400/10/2000	67 (37, 102)	0 (−30, 30)	<0.01 ${}^{†}$	30 (−21, 74)	15 (−23, 52)	0.47
400/10/2500	58 (10, 102)	−15 (−40, 20)	<0.01 ${}^{†}$	40 (−47, 88)	2 (−31, 61)	0.82
400/10/3000	85 (−19, 125)	12 (−14, 37)	0.02	80 (2, 125)	11 (−65, 85)	0.40
400/10/3500	23 (−13, 81)	40 (12, 80)	0.86	52 (10, 100)	17 (−31, 93)	0.91
**varying freq**						
400/10/2000	5 (−68, 53)	−80 (−110, −30)	0.02	−84 (−112, 2)	−33 (−89, −20)	0.65
400/12/2000	−50 (−84, 6)	−53 (−120, 31)	0.56	−161 (−210, −41)	−67 (−110, 14)	0.36
400/22/2000	50 (−152, 107)	−20 ( −46, −9)	0.39	−1 (−20, 82)	−10 (−194, 75)	0.13
400/30/2000	−20 ( −70, 10)	−72 (−108, −49)	0.03	−35 (−80, −4)	−86 (−140, 46)	0.47

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
