# Peer review of "Transthoracic Impedance Measured with Defibrillator Pads—New Interpretations of Signal Change Induced by Ventilations"

_jcm, 2019, doi:10.3390/jcm8050724_

Reviewer 1 Report

The authors have performed an interesting qualitative study evaluating if TTI peak corresponds with start of expiration in anesthetized patients. Although the results did not come up as expected, they still found that the maximum downslope of the ventilation TTI curve is time aligned with the start of expiration independently of other factors. Generally speaking, the study was well-designed, the results are adequately presented, and the discussion is appropriate. It is a very thoughtful paper and has the potential to add to the literature. I have only a few edits recommendation as below.

1. In Figure 1, the figure legends shown on the red square highlights of the figure, tTIpeak should be tTTIpeak, right?

2. In Figure 2, 3, and 4, the labels and the figure legends (eg, tpaw – tTIpeak) are squeezed together and hard to read. Again, tTIpeak is a typo of tTTIpeak?

Author Response

First of all, we would like to express our gratitude to the three reviewers for acknowledging the value of our work, and the encouraging comments on our experiments and results. Following the reviewers suggestions we have made changes to figures 1-4 and their captions to improve the readability of the figures and ease the interpretation of our results. The detailed response to each of the reviewers’ comments (which were minor in all cases) can be found following those remarks in blue color. The changes in the manuscript (captions in Figs2-4) appear in red in the revised manuscript.

The authors have performed an interesting qualitative study evaluating if TTI peak corresponds with start of expiration in anesthetized patients. Although the results did not come up as expected, they still found that the maximum downslope of the ventilation TTI curve is time aligned with the start of expiration independently of other factors. Generally speaking, the study was well-designed, the results are adequately presented, and the discussion is appropriate. It is a very thoughtful paper and has the potential to add to the literature. I have only a few edits recommendation as below.

1. In Figure 1, the figure legends shown on the red square highlights of the figure, tTIpeak should be tTTIpeak, right?

Yes, the reviewer is correct. We have corrected the typo in the Figure, and thank you for such a thorough review of the material.

2. In Figure 2, 3, and 4, the labels and the figure legends (eg, tpaw – tTIpeak) are squeezed together and hard to read. Again, tTIpeak is a typo of tTTIpeak?

The reviewer is right, we have redone the legends so that they are easy to read and put the two labels vertically aligned instead of horizontally aligned. And we have again corrected the tTTIpeak typo.

Reviewer 2 Report

Thank you for giving me an opportunity to review this manuscript, which reports a transthoracic impedance (TTI) measured with defibrillator pads. The aim of this study was to determine whether and how the insufflation phase could be precisely identified in the TTI. The authors synchronously measured TTI and airway pressure (Paw) in 21 consenting anaesthetized patients, TTI through the defibrillator pads and Paw by connecting the monitor-defibrillator’s pressure-line to the endotracheal tube filter. The authors concluded that the maximum downslope of the ventilation TTI curve is time aligned with the start of expiration independently of tidal volume, inspiratory time and ventilation frequency. Therefore, the authors addressed that this finding could be relevant in future investigations on the relationship between chest compressions and ventilations during CPR. As this manuscript is well written and is a very interesting one, I have no suggestions to be revised. Thank you again for their nice manuscript.

Author Response

First of all, we would like to express our gratitude to the three reviewers for acknowledging the value of our work, and the encouraging comments on our experiments and results. Following the reviewers suggestions we have made changes to figures 1-4 and their captions to improve the readability of the figures and ease the interpretation of our results. The detailed response to each of the reviewers’ comments (which were minor in all cases) can be found following those remarks in blue color. The changes in the manuscript (captions in Figs2-4) appear in red in the revised manuscript.

There were no requests from the reviewer so there are no specific answers to those comments.

Reviewer 3 Report

Thank you for submitting this interesting manuscript, I enjoyed reading about your study. I think the manuscript would be improved by some modifications from Fig 2-4. In particular, I think it would be helpful to use different symbols when plotting the TIpeak and maxDS data, and (even though they are defined elsewhere) define any abbreviations in the figure legend, such that the figure can stand alone to some extent.

Author Response

First of all, we would like to express our gratitude to the three reviewers for acknowledging the value of our work, and the encouraging comments on our experiments and results. Following the reviewers suggestions we have made changes to figures 1-4 and their captions to improve the readability of the figures and ease the interpretation of our results. The detailed response to each of the reviewers’ comments (which were minor in all cases) can be found following those remarks in blue color. The changes in the manuscript (captions in Figs2-4) appear in red in the revised manuscript.

Thank you for submitting this interesting manuscript, I enjoyed reading about your study. I think the manuscript would be improved by some modifications from Fig 2-4. In particular, I think it would be helpful to use different symbols when plotting the TIpeak and maxDS data, and (even though they are defined elsewhere) define any abbreviations in the figure legend, such that the figure can stand alone to some extent.

We have followed the reviewer’s advice and have used triangles for TTIpeak-maxDS instead of circles, and then we have also defined the abbreviations in the figure caption so that the figures are self-contained, these changes appear in red in the revised manuscript. As the reviewer can check the figure legends are now vertically aligned and with the fonts changed to improve readability, following the suggestions of reviewer 1.
